# Mitochondrial Processes during Early Development of *Dictyostelium discoideum*: From Bioenergetic to Proteomic Studies

**DOI:** 10.3390/genes12050638

**Published:** 2021-04-25

**Authors:** Monika Mazur, Daria Wojciechowska, Ewa Sitkiewicz, Agata Malinowska, Bianka Świderska, Hanna Kmita, Małgorzata Wojtkowska

**Affiliations:** 1Institute of Molecular Biology and Biotechnology, Faculty of Biology, Adam Mickiewicz University, 61-614 Poznan, Poland; monika.antoniewicz@gmail.com (M.M.); grobys.d@amu.edu.pl (D.W.); kmita@amu.edu.pl (H.K.); 2Department of Macromolecular Physics, Faculty of Physics, Adam Mickiewicz University, 61-614 Poznan, Poland; 3Institute of Biochemistry and Biophysics Polish Academy of Sciences, 02-106 Warszawa, Poland; ewa@ibb.waw.pl (E.S.); esme@ibb.waw.pl (A.M.); bianka.swiderska@gmail.com (B.Ś.)

**Keywords:** mitochondrial proteins, mass spectrometry, *Dictyostelium discoideum*, starvation, coupling efficiency, coupling capacity, spare respiratory capacity, qualitative and quantitative comparative studies

## Abstract

The slime mold *Dictyostelium discoideum’s* life cycle includes different unicellular and multicellular stages that provide a convenient model for research concerning intracellular and intercellular mechanisms influencing mitochondria’s structure and function. We aim to determine the differences between the mitochondria isolated from the slime mold regarding its early developmental stages induced by starvation, namely the unicellular (U), aggregation (A) and streams (S) stages, at the bioenergetic and proteome levels. We measured the oxygen consumption of intact cells using the Clarke electrode and observed a distinct decrease in mitochondrial coupling capacity for stage S cells and a decrease in mitochondrial coupling efficiency for stage A and S cells. We also found changes in spare respiratory capacity. We performed a wide comparative proteomic study. During the transition from the unicellular stage to the multicellular stage, important proteomic differences occurred in stages A and S relating to the proteins of the main mitochondrial functional groups, showing characteristic tendencies that could be associated with their ongoing adaptation to starvation following cell reprogramming during the switch to gluconeogenesis. We suggest that the main mitochondrial processes are downregulated during the early developmental stages, although this needs to be verified by extending analogous studies to the next slime mold life cycle stages.

## 1. Introduction

The unicellular slime mold *Dictyostelium discoideum*, one of the 150 species of *Dictyostelia* belonging to the Amoebozoa eukaryotic supergroup, displays the multicellularity form, namely aggregative or sorocarp multicellularity [1,2]. Due to starvation, the slime mold aggregates, which results in an intermediate migrating ‘slug’ stage and cells specializing into spores as well as three somatic cell types forming a stalk and structures to support the stalk and spore mass (Figure 1). This process is mainly based on chemical signals settled on cAMP and cAMP-dependent kinase (PkaC) [3], which induces the expression of the genes responsible for aggregation such as the receptor of cAMP (CarA), adenylate cyclase (AcaA), and cAMP phosphodiesterase (PdsA) [2,4]. cAMP pulses are initially secreted by a few starving cells, then cells move chemotactically toward the local cAMP and collect into molds, following which they form a slug and, finally, the fruiting body [5]. In addition, cAMP pulses upregulate the genes that are required during and after aggregation as tgrB and tgrC. These genes are responsible for cell adhesion and are known as the marker genes of post-aggregative cell differentiation [6].

*D. discoideum* cells, during the early steps of aggregation when unicellular cells start to move and form waves, are referred to as early developmental stages induced by starvation (Figure 1) [1]. These early developmental stages have been studied previously [7,8]. Accordingly, transcriptomic and cell proteomic data on mutant strains lacking genes that are crucial for cell aggregation are available [7]. The data revealed the downregulation of mitochondria proteins of the Krebs cycle and oxidative phosphorylation (OXPHOS). Another application of comparative MitoProteome studies for mitochondria isolated from different developmental stages including the early ones, is the general downregulation of crucial mitochondrial proteins and the upregulation of alternative oxidation (AOX) [8] during the early stages. The results of these studies were related to the mitochondrial energy status that was detected in vivo with a MitoTracker. However, these studies were not functionally based, which could have provided more precise answers to the question regarding mitochondrial functionality during starvation-induced aggregation.

Undoubtedly, *D. discoideum* is a popular model organism for studying cellular processes such as cell signaling, cell differentiation, and morphogenesis [9], the evolution of sociality and prey–predator relationships [3]. Due to the wide range of human orthologue genes, slime mold can be used to study human diseases, particularly neurodegeneration [10]. Moreover, *D. discoideum* is also a perfect model for research concerning intracellular and intercellular mechanisms such as early and late aggregation from unicellularity to multicellularity, in which mitochondria seem to have a crucial function [4].

It has been shown that cell differentiation is under the control of the nutrient glucose [11,12,13]. In our study, we performed an estimation of mitochondria’s functional state in intact cells representing the unicellular (U), aggregation (A), and streams (S) stages and combined the data with comparative proteomic analysis. The obtained results indicate that the observed differences among the studied stages may be associated with ongoing adaptation to starvation following the cell reprogramming that is supported by the metabolic background of starving cells. To our knowledge, this is the first report presenting combined *D. discoideum* mitochondrial proteomics and bioenergetic studies.

## 2. Methods

### 2.1. Cell Culture and Mitochondria Isolation

*D. discoideum* AX2 cell cultures were performed according to [14]. For the measurement of oxygen uptake using the Clarke electrode (Section 2.3), cells were cultured exponentially until they reached a density of 2.0–3.0 × 10^6^ cells/mL (the doubling time of 10–11 h) in an axenic growth medium (1% *w/v* protease peptone, 0.5% *w/v* yeast extract, 0.45% *w/v* D-glucose, 0.09% *w/v* Na_2_HPO_4_, and 0.23% *w/v* KH_2_PO_4_). Cells were then cultured in 100 mL at 19 °C using a Certomat MO rotary shaker at 90 rpm. The cells of stage U were transferred to sterile 24-well plates at 1 million cells per well and left for 1 h for cell adhesion. Cells at stages A and S were obtained by washing the cell culture twice in the developmental buffer (DB) starvation medium containing 5 mM of Na_2_HPO_4_, 5 mM of KH_2_PO_4_, 1 mM of CaCl_2_, and 2 mM of MgCl_2_, with a pH of 6.5 [15]. The cells were then transferred to the sterile 24-well plates for 23 h for the A stage and 25 h for the S stage cells. The presence of the U, A, and S stages was confirmed by using a Nikon Eclipse TE 2000-U microscope. For mitochondria isolation, cell cultures (850 mL)at a given stage were centrifuged at 600× *g* for 5 min and then washed twice in phosphate medium A (14.5 mM KH_2_PO_4_, 5 mM Na_2_HPO_4_, pH 6) by centrifugation at 600× *g* for 5 min. Next, the cells were homogenized in medium B (0.38 M sucrose, 20 mM Tris-Cl, 0.5 mM EDTA, 1% defatted bovine serum albumin [BSA], pH 7.5) using a glass/teflon homogenizer and centrifuged at 860× *g* for 5 min. After this step, the supernatant was centrifuged at 10,000× *g* for 15 min and the obtained pellet was suspended in medium C (0.38 M sucrose, 20 mM TrisCl, 0.5 mM EDTA, 0.6% defatted BSA, pH 7.2), and centrifuged at 860× *g* for 5 min to remove debris. The supernatant was centrifuged at 10,000× *g* for 10 min, the mitochondrial pellet was suspended in medium D (0.38 M sucrose; 20 mM, Tris-Cl, pH 7.2), and centrifuged at 10,000× *g* for 15 min. The mitochondrial pellet was suspended in medium D and used for further analysis. All centrifugation procedures were carried out at 4 °C and media used for mitochondria isolation were ice cold. The calculated mean values of the yield of mitochondria for 1 g of cells were 5 mg, 1.4 mg, and 1.7 mg for stage U, A, and S, respectively.

### 2.2. The Measurements of Cell Respiration and the Status of Mitochondrial Coupling in Intact D. discoideum Cells

To measure the oxygen consumption rates of *D. discoideum,* cells from the U, A, and S stages collected from the 24-well plates were centrifuged (11,000× *g*, 3 min) and resuspended in the DB containing 5 mM of Na_2_HPO_4_, 5 mM of KH_2_PO_4_, 1 mM of CaCl_2_, and 2 mM of MgCl_2_ at a pH of 6.5 [15]. The rate of the cells’ oxygen uptake (2 × 10^6^) was measured in 0.5 mL of the DB in a water-thermostated incubation chamber with a computer-controlled Clark-type O_2_ electrode at 18 °C (Oxygraph, Hansatech, UK). To estimate the functional states of the mitochondria at the studied stages, mitochondrial coupling capacity and mitochondrial coupling efficiency were calculated based on basal (actual) respiration, state 4 (the resting state), state 3 (the phosphorylating state), and maximal respiration. Basal cell respiration was measured in the DB with supplementation of pyruvate at a final concentration of 10 mM. State 4 was enforced through the addition of 0.5–1.5 µM of tributyltin (TBT), which is an inhibitor of mitochondrial ATP synthase, whereas maximal respiration was induced by using 0.1–0.24 µM of carbonyl cyanide-p-trifluoromethoxyphenylhydrazone, an uncoupler (FCCP). State 3 was calculated by subtracting state 4 from basal respiration. The contribution of state 3 to the basal respiration corresponded to mitochondrial coupling efficiency, while the mitochondrial coupling capacity that corresponded to FCCP uncoupling capacity was estimated as the maximal respiration to state 4 ratio. The TBT and FCCP concentrations were titrated for each stage [16,17,18]. To estimate the ability of the substrate supply and electron transport to respond to an increase in energy demand, mitochondrial spare respiratory capacity was estimated through the determination of the difference between maximal respiration and basal respiration [19].

### 2.3. Sample Preparation for Mass Spectrometry Analysis

Mass spectrometry experiments were performed at the Mass Spectrometry Laboratory at the Institute of Biochemistry and Biophysics, PAS. A total of 5 µg of isolated mitochondria was suspended in a dissolution buffer (8 M of urea, 2% CHAPS, 10 mM of HEPES, with a pH of 8.5). Samples were processed using the modified single-pot solid-phase-enhanced sample preparation (SP3) method [20]. The exact sample preparation protocol was as follows. Cysteines were reduced through 1 h of incubation with 20 mM of tris (2-carboxyethyl)phosphine (TCEP) at 37 °C, followed by 10 min of incubation at room temperature with 50 mM of methyl methanethiosulfonate (MMTS). The magnetic bead mix was prepared by combining equal parts of Sera-Mag Carboxyl hydrophilic and hydrophobic particles (09-981-121 and 09-981-123, GE Healthcare Lifesciences, Upsala, Sweden). The bead mix was washed three times with MS-grade water and resuspended in a working concentration of 10 µg/µL. Furthermore, 8 µL of the prepared bead mix, along with 5 µL of 10% formic acid and 500 µL of acetonitrile, was added to each sample. The proteins that bound to the beads were washed with 75% ethanol, isopropanol, and acetonitrile, followed by overnight digestion with 1 µg of trypsin/Lys-C mix (Promega). After digestion, the peptides were washed with acetonitrile and eluted from the beads by subsequent incubation with MS-grade water and 2% DMSO, with sonication during each step. The pulled aliquots were dried in a SpeedVac and resuspended in 60 µL of 2% acetonitrile and 0.1% formic acid. Peptide concentrations were measured using the Pierce Quantitative Colorimetric Peptide assay (Thermo Scientific, Rockford, IL, USA).

### 2.4. Mass Spectrometry

Three µg of digested samples were analyzed using the LC–MS system comprising a high-performance liquid chromatography (UPLC) chromatograph (nanoAcquity, Waters) and a Q Exactive mass spectrometer (Thermo). Peptides were trapped on a C18 pre-column (180 µm × 20 mm, Waters) using a 0.1% water solution of FA as a mobile phase then transferred to a BEH C18 column (75 µm × 250 mm, 1.7 µm, Waters) using an ACN gradient (0–35% ACN in 160 min) in the presence of 0.1% FA at a flow rate of 250 nL/min. The spectrometer was working in a data-dependent mode. To prevent cross-contamination, blank runs were performed between the sample runs.

### 2.5. Data Analysis

The obtained data were pre-processed with Mascot Distiller software (Matrixscience), and protein identification was performed using Mascot Server 2.5 (Matrixscience) against the *D. discoideum* protein sequences (13921 sequences) deposited in the NCBInr database (20180903, 167148673 sequences; 60963227986 residues). The parameters were set as follows: enzyme—trypsin, missed cleavages—1, fixed modifications—methylthio (C), variable modifications—GlyGly (K), oxidation (M), and instrument—HCD. To reduce mass errors, peptide and fragment mass tolerance settings were established separately for each file after an off-line mass recalibration [21]. The assessment of confidence was based on the target/decoy database search strategy [22], which provided *q*-value estimates for each peptide spectrum match. All queries with a *q*-value of >0.01, subset proteins, and proteins identified with one peptide were discarded from further analysis. The mass recalibration, false discovery rate (FDR) computations, and data filtering were done with Mscan software that was developed in-house (MScan 2.0.3 (accessed on 26 June 2018)).

The lists of identified peptides were merged and overlaid onto 2-D heatmaps generated from the LC-MS spectra, and the volumes were obtained from the assigned peaks (a more detailed description of data extraction procedures can be found in [23]). Quantitative values were then exported into text files for statistical analysis with Diffprot [21] software. The calculated *p*-values were adjusted for multiple testing using a procedure that controlled for a FDR. Diffprot was run with the following parameters: the number of random peptide sets = 10^6^, clustering of peptide sets—only when 90% were identical, normalization through LOWESS, and quantification based on unique peptides. Only proteins with a *q*-value below 0.05 or those present in only one of the two compared analytical groups were taken into consideration during further analysis. The tendency in comparative studies as down in A and up in S was calculated from the ratio values of comparative qualitative studies for phase to phase ratio quotient U/A and A/S (Table 1). The expression profile was calculated against the U state as a basal of expression level (1). The ratio value (>1) and (<1) was regarded as upregulated and downregulated, respectively. The phase to phase ratio quotient “[S/U]” was taken for the proteins isocitrate dehydrogenase (NAD+), cytochrome c oxidase subunit IV, and heat shock protein Hsp70 family protein. For other proteins, the [S/U] was calculated from the equation [A/U]/[A/S], where [A/U] and [A/S] are given as the ratio in Section for “Phase to phase ratio quotient [A/U]” and Section for “Phase to phase ratio quotient [A/S]”.

## 3. Results and Discussion

### 3.1. The Functional Status of Intact D. discoideum Cells’ Mitochondria at the Early Developmental Stages

The respiratory state of mitochondria is a dynamic state resulting from crosstalk between the two bioenergetic states termed state 3 and state 4 [16,19]. State 4, also referred to as the proton-selective leak, is characterized by low oxygen uptake and higher inner membrane potential, while state 3, also referred to as the phosphorylating state due to resulting ADP phosphorylation to ATP, reflects high oxygen uptake and lower inner membrane potential. Maximal respiration is triggered in the presence of an uncoupler and represents the respiratory chain’s ability to support additional energetic demands [16,19]. All of the states are crucial for the estimation and comparison of the functional status of different intact cells’ mitochondria, as they allow for the calculation of the following parameters: mitochondrial coupling efficiency, mitochondrial coupling capacity, and mitochondrial spare respiratory capacity, as described in the Methods (Section 2.2). The parameter calculation allows for the comparison of cells containing different amounts of mitochondria, which was observed herein (Methods, Section 2.5) and has already been reported for cell in stages U, A, and S [8]. The representative traces registered for cells in stages U, A, and S are presented in the Figure 2A whereas raw data used for the calculation of all bioenergetic parameters are presented in Appendix A. As shown in Figure 2B, intact *D. discoideum* cells in stages U, A, and S displayed the same rate of state 4 respiration (i.e., the proton leak), but differed in their basal, state 3, and maximal respiration rates. The values of the parameters were higher for cells in the U stage than for cells in the A and S stages, while the cells in stages A and S did not differ in the values of their parameters, except for maximal respiration, which was lower for cells in the S stage. However, according to the yield of mitochondria isolation, it could be assumed that cells in stages A and S contained the comparable mitochondria mass that was lower than in cells in the U stage.

Basal respiration is usually strongly controlled by ATP turnover; hence, it alters in response to ATP demand [19]. The observed differences in basal respiration rates co-occurred with a decrease in the rate of state 3, suggesting that the cells in stages A and S displayed a lower demand for ATP (Figure 2B). Accordingly, mitochondrial coupling efficiency corresponding to state 3 respiration contribution to basal respiration decreased to the same level for the cells in stages A and S when compared to the cells in stage U (Figure 2C. Thus, the cells in stages A and S displayed lower levels of ATP synthesis than those in stage U. However, the cells in stages A and S differed in the uncoupling capacity of FCCP (Figure 2B), which resulted in differences in mitochondrial coupling capacity (Figure 2D), defined as the ratio of maximal respiration to state 4 respiration. The value of the parameter is sensitive to changes in substrate oxidation and proton leakage [19]. Accordingly, FCCP-imposed uncoupling should reflect the maximum activity of electron transport and substrate oxidation, which are achievable by cells under the given conditions. Therefore, we compared the mitochondrial spare respiratory capacity of the studied cells. The parameter corresponds to the difference between maximal respiration and basal respiration and reflects how close to its bioenergetic limit a cell is operating [19]. One of the known factors that influence the extent of mitochondrial spare respiratory capacity is substrate availability, which is based on substrate entry into the Krebs cycle that is synchronized with the electron transport of the respiratory chain. As shown in Figure 2E, the cells in stage S displayed a distinct decrease in mitochondrial spare respiratory capacity when compared to the cells in stages U and A, but the value of the parameter increased for the cells in stage A when compared to the cells in stage U. Interestingly, it has been shown that mitochondrial spare respiratory capacity increases in starvation-resistant cells [24], which may permit rapid adaptation to metabolic changes [25]. On the other hand, cell reprogramming may correlate with a decrease in mitochondrial spare respiratory capacity [26], which could explain the distinct decrease in the parameter value of the cells in stage S [27]. Overall, the analysis of bioenergetic parameters indicated decreased ATP synthesis during stages A and S, which co-occurred in the case of the latter, with the limited access of substrates to the respiratory chain. These changes may be associated with an ongoing adaptation to starvation, followed by cell reprogramming. Therefore, we decided to study the changes in mitochondrial proteome differences among the cells in stages U, A, and S.

### 3.2. Quantitative Comparative Analysis of the Mitochondrial Proteomes of the Early Developmental Stages

In mitochondria isolated from the cells in stages U, A, and S, and due to the mass spectrometry analysis, we detected 294 proteins overall (Appendix A and segregated them into nine functional groups of proteins: protein synthesis and degradation, signaling, protein import, substrate transport, mitochondrial gene expression, the Krebs cycle, OXPHOS, other known processes (fatty acid metabolism, redox), and uncharacterized proteins (Figure 3).

The numbers of proteins assigned to the mentioned functional groups were comparable for the studied mitochondria. As presented in Figure 3, the most abundant groups were protein biosynthesis and degradation, OXPHOS, and other known processes. The less-abundant groups were transport, protein import, the Krebs cycle, and uncharacterized proteins. The least-numerous groups of proteins were the signaling and gene expression functional groups. Thus, the mitochondrial proteomes obtained for the studied early developmental stages were comparable. Therefore, we decided to perform a qualitative comparative analysis of the obtained mitochondrial proteome data.

### 3.3. Qualitative Comparative Analysis of the Mitochondrial Proteomes of the Early Developmental Stages

Comparative qualitative analyses of the detected proteins were performed for the following pairs of stages: A/U, S/U, and A/S regarding the previously described functional groups of proteins. Analyses were done according to the procedure described in the Methods section (Section 2.3, Section 2.4 and Section 2.5).

#### 3.3.1. The Krebs Cycle and OXPHOS

The Krebs cycle and OXPHOS are well-known, crucial mitochondrial processes of energy transformation. The comparative analysis revealed the presence of four proteins of the Krebs cycle (citrate synthase (XP_643860.1), isocitrate dehydrogenase (NAD^+^) (XP_628920.1), succinate-CoA ligase (XP_636263.1), and malate dehydrogenase (XP_629516.1) and two proteins functionally linked to this cycle (3-oxoacid CoA-transferase and phosphoenolpyruvate carboxykinase) (XP_636911.1) (Table 1). We observed decreased levels in the majority of the mentioned proteins in stage A and increased levels in stage S, termed the ‘down in A and up in S tendency’ (Table 1), which often resulted in down- or upregulation in stage S when compared to stage U (Figure 4A,B).

As shown in Figure 4A, when the detected Krebs cycle proteins were compared between stages S and U, succinate-CoA ligase was upregulated, citrate synthase and isocitrate dehydrogenase (NAD^+^) were downregulated and malate dehydrogenase returned to stage U level, which denoted no change in its expression profile. The mitochondrial malate dehydrogenase catalyzed the reaction of the malate-to-oxaloacetate conversion with the use of NAD^+^ reduction to NADH [28,29]. It has been shown that enzyme activity is inhibited by differentiation-inducing-factor 1 (DIF-1), which affects cell energy transformation and leads to the inhibition of cell proliferation. These data suggest that the described inhibition of malate dehydrogenase by DIF-1 could be one of the mechanisms that induce the anti-proliferative effects that remain in agreement with the differentiation processes that start to occur at stage A. However, the steady state of its expression profile seemed not to influence the Krebs cycle.

The tendency of ‘down in A and up in S’ was also observed for the two enzymes functionally linked to the Krebs cycle. The cycle 3-oxoacid CoA-transferase was finally downregulated in stage S when compared to stage U (Figure 4B). Interestingly, the product of the human 3-oxoacid CoA-transferase (OXCT1) was converted to acetyl-CoA and finally fed into the Krebs cycle [30]. Mitochondrial phosphoenolpyruvate carboxykinase (PCK2) (XP_645396.1) [31] is a gluconeogenesis enzyme that catalyzes the phosphorylation reaction of oxaloacetate. PCK2 was the enzyme that was finally upregulated in stage S when compared to stage U (Figure 4A). We assumed that starving *D. discoideum* cells in stages A and S caused glucose deficiency that could trigger gluconeogenesis [32]. Interestingly, in our study, we observed an increase in phosphoenolpyruvate carboxykinase (PCK2) in stage S. The decrease in Krebs-cycle enzyme expression in stage A may coincide with enhanced gluconeogenesis, which is used to produce glucose during starvation. Unexpectedly, in our comparative analysis, we did not observe a significant change in proteins in the OXPHOS functional group, except for the cytochrome c-oxidase, which was downregulated (Figure 4C).

Our obtained data coincided with the low state 3 contributions to basal respiration in stages A and S, reflecting decreased OXPHOS, decreased spare respiratory capacity in stage S, and possibly enhanced gluconeogenesis. Nevertheless, the obtained data did not reveal an upregulation of any glycolytic enzymes that could have explained the aerobic glycolysis that is regarded as typical for proliferative cells, similar to cancer [33]. This is because the study was focused on mitochondrial proteins. However, glycolytic enzymes could interact with mitochondria, as glycolysis occurs in the cytosol.

The data indicating the downregulation of the Krebs cycle enzymes coincide with the results that were obtained in [7,8]. We concluded that the main function of the mitochondria in the early developmental stages emerged temporarily, rewriting the mitochondrial metabolism from the Krebs cycle into gluconeogenesis. We showed, for the first time, the presence of PCK2 upregulation during *D. discoideum* starvation. The regulation of PCK2 at the level of the transcription of the encoding gene is under the positive control of cAMP and is crucial for cell aggregation. Indeed, the deficiency of glucose during starvation activates adenylate cyclase, which actives CREB and, finally, PCK2. This could take place when phosphoenolpyruvate (PEP) cannot be converted into pyruvate, which is used in the gluconeogenesis pathway (Figure 4C) [31].

#### 3.3.2. Protein Metabolism

In the case of the protein biosynthesis and degradation group, the obtained data revealed three enzymes that were responsible for the metabolism of mitochondrial amino acids (Table 1). The general tendency concerning their expression level remained ‘down in A and up in S’ and was observed for acetyl glutamate kinase (XP_637813.1), 4-aminobutyrate transaminase (XP_647552.1) (Figure 4A), and branched-chain amino acid aminotransferase (XP_638096.1). The latter did not return to the level of stage U, so the enzyme was classified as downregulated in stage S when compared to stage U (Figure 4B). Acetyl glutamate kinase is a NAD^+^ binding enzyme engaged in arginine biosynthesis [28]. In *D. discoideum*, the ArgC enzyme is similar to the bifunctional acetyl glutamate kinase and N-acetyl-γ-glutamyl-phosphate reductase from yeast, which catalyze the second and third steps in the biosynthesis of the arginine precursor ornithine. The detection of this protein revealed the known connection between the urea and Krebs cycles. However, we did not observe any changes in protein expression in the case of the urea cycle. The 4-aminobutyrate transaminase catalyzed the conversion of 4-aminobutanoate and 2-oxoglutarate into succinate semialdehyde and L-glutamate [33]. This process is dedicated to glutamate degradation and is also correlated with the urea and Krebs cycles. The expression profile of these proteins included downregulation in A and an increase in S, resulting in upregulation (Figure 4A). In the case of the branched-chain amino acid aminotransferase, another example of aminotransferase, the transamination of branch-amino acids as leucine, isoleucine, and valine as well as their respective alfa-ketoacids, was reported [28]. For this protein, an opposite decreasing tendency was observed, so we classified the protein as downregulated in stage S compared to stage U (Figure 4B). We speculate that the amino acids’ biosynthesis protein expression profile revealed the metabolic flexibility of *D. discoideum* cells determined by their ability to reprogram anabolic and catabolic pathways, presumably through altering gene expression programs and intercellular interactions within the glucose-free microenvironment during starvation [34]. This reprogramming could be based on the temporary shift from the Krebs cycle to gluconeogenesis, with amino acids as the fuel for gluconeogenesis (Figure 4C).

The comparative study on the discussed groups of proteins also revealed that the presence of cytosolic chaperone Hsp70 (XP_643155.1) significantly increased in stage A when compared to stage U and slightly decreased in stage S when compared to stage U (Figure 4A). Hsp70 is a chaperone that plays a crucial role in protein folding, disaggregation, and degradation [35]. It was also shown that ribosomes stay in close proximity to mitochondria [36], which implies their co-translational import into mitochondria [37]. Therefore, the presence of the mentioned chaperone could ensure sufficient protein refolding, which is essential for the successful import of protein into mitochondria.

To summarize, the detected changes in the expression levels of the functional protein group suggest that at the early developmental stages, the processes of mitochondrial amino acid metabolism are upregulated, which we showed here for the first time.

#### 3.3.3. The Import of Proteins into Mitochondria and the Transport of Other Molecules

Most mitochondrial proteins are synthesized in the cytosol and have to be delivered to the various mitochondrial sub-locations. This process is regulated by the presence and activity of import apparatus consisting of proteins that, in general, recognize and translocate proteins across mitochondrial membranes (for details, see [37]). In this comparative study, we detected mtHsp70 (XP_629204.1) [38], which showed the ‘down in A and up in S’ expression profile tendency. We estimated that mtHsp70 was upregulated in stage S when compared to stage U (Figure 4A). This mitochondrial chaperone is responsible for the protein assembly of proteins delivered into the mitochondrial matrix and acts as a subunit of the PAM import machinery that is crucial for the proper unidirectional transfer of precursor proteins across the inner mitochondrial membrane. It seems that this process could be upregulated in stage S. The elevated level of cytosolic Hsp70 could be arranged in the same processes as those involved when the chaperone assists in the partial refolding of precursor proteins for import across the protein channel [8]. Moreover, mtHp70 is a chaperone that plays an important role in iron–sulfur cluster (IS) biogenesis [39]. Studies on yeast and human ISC involve gene mutations to show genome instability and the induction of the DNA damage repair pathway [40]. This, in turn, indicates that pathways related to ISC biosynthesis are strongly conserved—more so than for ATP production, for example—among diverse mitochondrial homologues [41]. This also explains ISC’s presence (as only one common module to mitochondria) in mitosomes. This suggests that the essential process of ISC biosynthesis occurs in stage S.

In the case of the functional transport protein group, we detected an uncharacterized mitochondria substrate carrier protein named mt carrier Q (XP_645160.1). Its level significantly increased in stage A when compared to stage U and decreased in stage S when compared to stage U, although its level was higher when in stage U (Figure 4A). We conclude that the organization of the mitochondrial import machineries do not undergo major changes during the early developmental stages. We identified a new, uncharacterized transport protein, which was significantly upregulated in stage S (Figure 4A) and is worth further study to discern its exact function in the process of aggregation.

#### 3.3.4. Other Known Processes

The functional group, termed ‘other known processes’, included enzymes engaged in fatty acid metabolism and redox processes. The oxidation of fatty acids may provide another source for biosynthetic growth under glucose limitation during *D. discoideum* starvation [42]. Our comparative study provided four enzymes of the fatty acid metabolism acetyl-CoA C-acetyltransferase (XP_645587.1), which showed the ‘down in A and up in S’ tendency, but its level in stage S was lower than in stage U, denoting a final downregulation (Table 1 and Figure 4B). The ‘down in A and up in S’ tendency was also observed for the electron transfer of flavoprotein α and β subunits (XP_635485.1 and XP_642058.1). However, the comparison of the proteins’ levels between stages U and S indicate their upregulation (Figure 4A). Interestingly, acyl-CoA oxidase A (XP_643323.1) engaged in β-oxidation that was elevated in stage A when compared to stage U and strongly decreased in stage S when compared to stage U (Figure 4B). This profile was distinctly different from that observed for the previous groups of proteins. Therefore, we can assume the diminution of β-oxidation.

During β-oxidation, acetyl-CoA is produced, then enters the Krebs cycle and consequently stimulates ATP synthesis. However, during starvation, acetyl-CoA could activate the first enzyme of gluconeogenesis—the pyruvate carboxykinase that converts pyruvate into oxaloacetate. As shown, the detected β-oxidation proteins were mostly downregulated in stage S when compared to stage U. Therefore, *D. discoideum* cells probably do not use acetyl-CoA during starvation to stimulate gluconeogenesis (the first enzyme of this pathway). Consequently, glycerol is not used as a carbon source for *D. discoideum’s* gluconeogenesis. Thus, as described, during starvation, gluconeogenesis may start due to PCK2, which has been proven to upregulate in stage S when compared to stage U and uses amino acids as fuel. Thus, amino acids are metabolized into the oxaloacetate, then converted into PEP by mitochondrial PCK2 (Figure 4C) [43].

Within the described group of other known processes, proteins engaged in the redox reaction were also enclosed. For this group, we detected mitochondrial superoxidase dismutase (SOD2) (XP_645815.1). SOD2 followed the ‘down in A and up in S’ pattern. It was finally upregulated in stage S when compared to stage U, which coincides with the available data [8]. This upregulation was not very significant (Figure 4A) but remained in agreement with the data concerning the role of ROSs as signaling molecules during aggregation, where SOD control signaling molecules are essential for proper development [44]. The ROS production could increase as a response of mitochondrial metabolism reprogramming. Thus, additional studies in this field are needed.

#### 3.3.5. Uncharacterized Proteins

Two uncharacterized mitochondrial proteins were detected in the comparative analysis (Table 1). The putative mitochondrial transferase caf17 (XP_639996.1) and an uncharacterized protein (XP_639145.1), named in the *D.discoideum* database (dictybase) ‘as putative delta-1-pyrroline-5-carboxylate dehydrogenase’, were discovered. Both followed the ‘down in A and up in S’ tendency. The putative mitochondrial transferase caf17 was presumably involved in the incorporation of iron–sulfur clusters (ISC) into mitochondrial aconitase-type proteins. ISC was previously mentioned as being a common but crucial molecule present in mitochondria and related to organelles [45], which is thought to reflect its basic mitochondrial function. The ‘putative delta-1-pyrroline-5-carboxylate dehydrogenase’ is probably involved in synthesis of L-glutamate from L-proline, suggesting its function in the amino acid degradation process (Figure 4C). Both uncharacterized proteins were upregulated in stage S when compared to stage U (Figure 4A) and warrant a deeper investigation.

## 4. Conclusions

Our study on *D. discoideum* cells in early developmental stages revealed differences in their mitochondria functional state. The differences correspond to the comparative proteomic analysis results and may be associated with diminution of β oxidation, Krebs cycle, and oxidative phosphorylation, and enhancement of the metabolism of amino acids, being a known fuel for gluconeogenesis during starvation. This in turn may support metabolically ongoing adaptation to starvation and following cell reprogramming.

## Figures and Tables

**Figure 1 genes-12-00638-f001:**
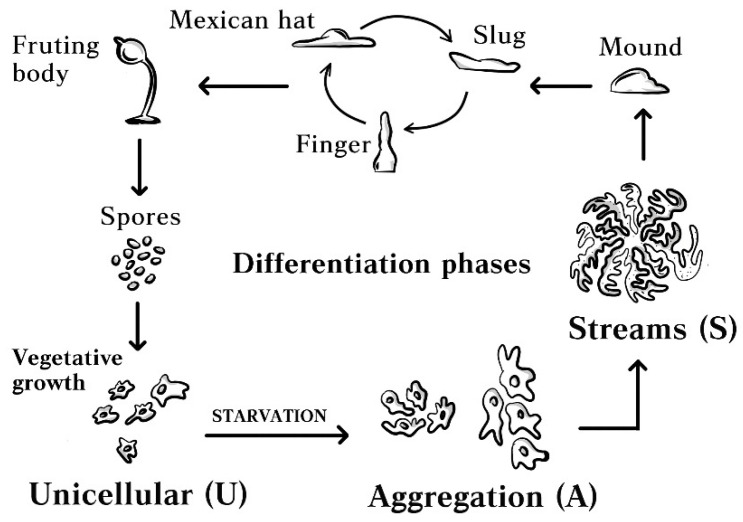
The life cycle of *D. discoideum.* The studied stages were marked by bold letters.

**Figure 2 genes-12-00638-f002:**
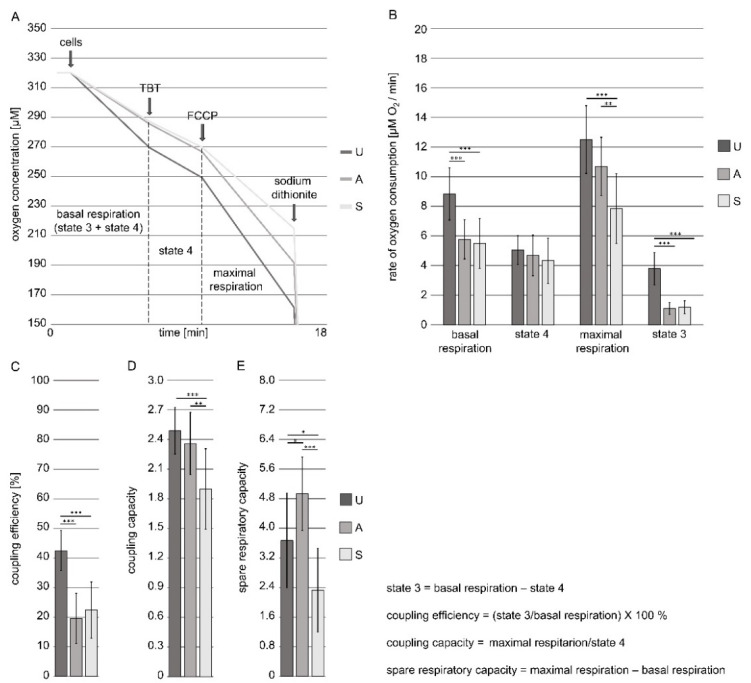
The functional state of mitochondria in intact *D. discoideum* cells in the studied stages. (**A**) Representative traces for cells in studied stages applied for calculations as well as explanation of the calculated parameters; (**B**) oxygen consumption rates under the applied oxygen uptake rate measurement condition; (**C**) coupling efficiency (contribution of state 3 to basal respiration); (**D**) coupling capacity (maximal respiration to state 4 ratio); (**E**) spare respiratory capacity (the difference between maximal respiration and basal respiration). The rate of oxygen uptake by cells (2 × 10^6^) was measured in 0.5 mL of DB in a water-thermostated incubation chamber with a computer-controlled Clark-type O_2_ electrode at 18 °C (Oxygraph, Hansatech, UK). Tributyltin (TBT) was added in a concentration (0.5–1.5 µM) to enforce state 4. The maximal respiration was induced by using 0.1–0.24 µM of carbonyl cyanidep-trifluoromethoxyphenylhydrazone (FCCP); sodium dithionite is a zero oxygen solution powder used for calibration of oxygen sensors at zero oxygen. The data are presented as mean values ± SD of three independent experiments. * *p* < 0.05; ** *p* < 0.005, *** *p* < 0.001. Calculated original data are presented in Appendix A.

**Figure 3 genes-12-00638-f003:**
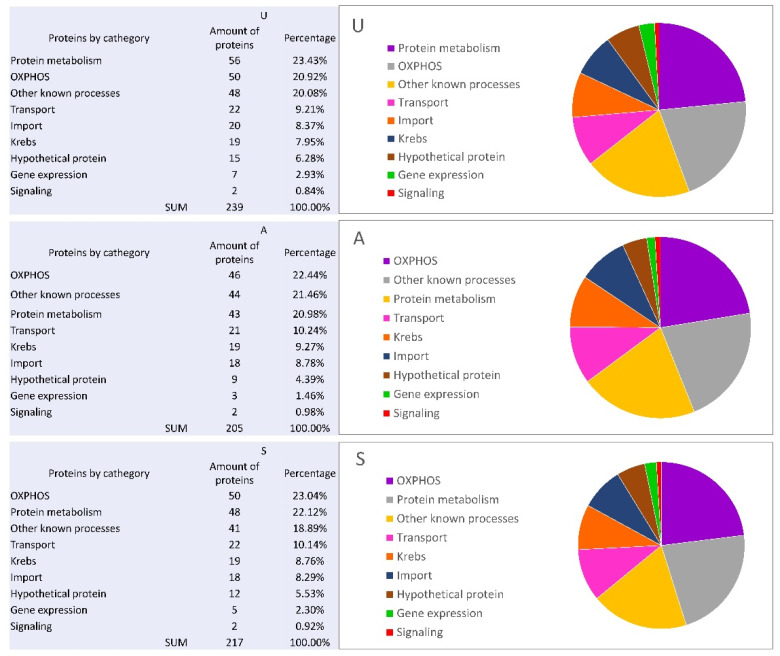
The numbers of *D. discoideum* mitochondrial proteins assigned to the nine functional groups. The classification of identified proteins was performed at the base of the dictybase.org and proteins belonging to the same functional groups were counted in order to assess the size of the classes for details (see Methods Section 2.5 and Appendix A).

**Figure 4 genes-12-00638-f004:**
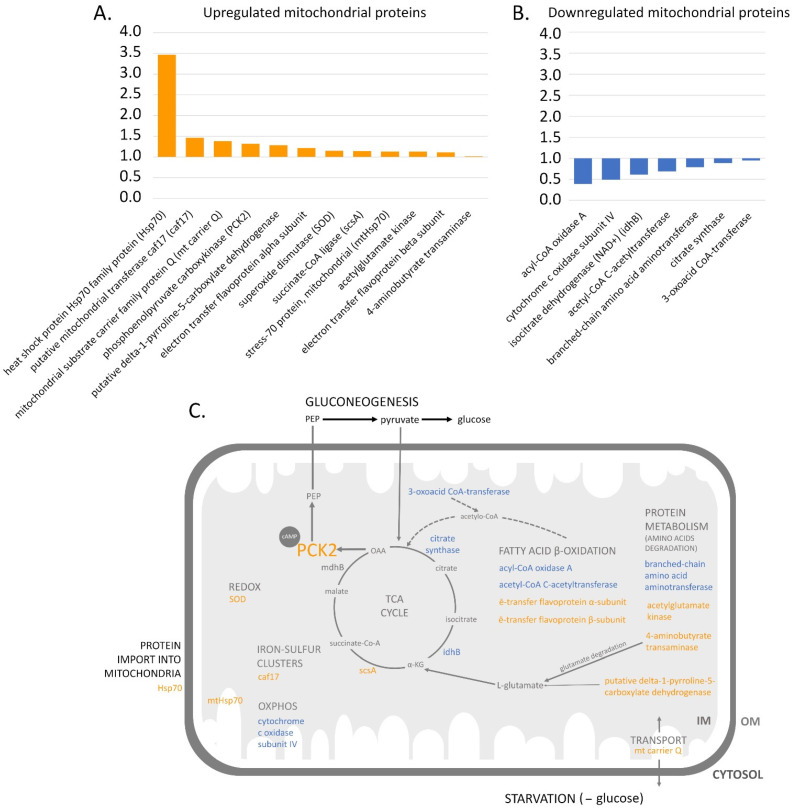
The mitochondrial proteins upregulated and downregulated in stage S when compared to stage U during *D. discoideum* starvation. (**A**) Upregulated mitochondrial proteins; (**B**) downregulated mitochondrial proteins. The protein upregulation and downregulation were estimated using qualitative comparative analysis between stages U and S (see Methods Section 2.5). (**C**) Diagram of downregulated (blue) and upregulated (orange) mitochondrial proteins assigned to cellular processes. The cell metabolic adaptation under glucose depletion includes attenuation of the Krebs cycle, fatty acid β oxidation, and oxidative phosphorylation, and enhancement of the amino acids’ metabolism, being known as a fuel for gluconeogenesis. The early steps of gluconeogenesis is regulated by mitochondrial phosphoenolpyruvate carboxykinase (PCK2), which allows for alternative glucose/biosynthesis intermediates required for cell aggregation. TCA (tricarboxylic acid cycle, Krebs cycle); PEP (phosphoenolopyruvate); SOD (superoxide dismutase); caf17 (putative mitochondrial transferase); scsA (succinate-CoA ligase); mdhB (mitochondrial malate dehydrogenase); idhB (isocitrate dehydrogenase (NAD+)); α-KG (α-ketoglutarate); OM (mitochondrial outer membrane); IM (mitochondrial inner membrane).

**Table 1 genes-12-00638-t001:** Results of the comparative mitoproteomes of the U (unicellular), A (aggregation), and S (streams) stages. *D. discoideum* stages regarding the distinguished functional groups: A/U, S/U, and A/S.

RefSeq Database	Protein Name	Phase to Phase Ratio Quotient A/U				Phase to Phase Ratio Quotient S/U				Phase to Phase Ratio Quotient A/S			
		*q* Value	Ratio	Fold Change	Peptides	*q* Value	Ratio	Fold Change	Peptides	*q* Value	Ratio	Fold Change	Peptides
	Krebs												
XP_643860.1	citrate synthase, mitochondrial	0.00073	0.56	1.78	46					0.00023	0.63	1.6	45
XP_628920.1	isocitrate dehydrogenase (NAD+) (IdhB)	0.04842	0.66	1.52	25	0.01253	0.61	1.64	25				
XP_636263.1	succinate-CoA ligase (scsA)	0.00858	0.48	2.1	16					0.00599	0.42	2.37	16
XP_629516.1	malate dehydrogenase (mdhB)	0.00027	0.58	1.71	45					0.00006	0.58	1.73	45
	-												
XP_636911.1	3-oxoacid CoA-transferase	0.00569	0.55	1.82	29					0.04059	0.58	1.73	29
XP_645396.1	phosphoenolpyruvate carboxykinase (PCK2)	0.04045	0.54	1.86	30					0.00055	0.41	2.43	32
	OXPHOS												
XP_640649.1	cytochrome c oxidase subunit IV					0.03247	0.49	2.06	19				
	Protein biosynthesis												
XP_643155.1	heat shock protein Hsp70 family protein	0.00015	3.76	3.76	32	0.00007	3.47	3.47	33				
XP_637813.1	acetylglutamate kinase	0.00886	0.61	1.65	37					0.00061	0.54	1.86	37
XP_638096.1	branched-chain amino acid aminotransferase	0.00015	0.45	2.21	22					0.01041	0.57	1.75	21
XP_647552.1	4-aminobutyrate transaminase	0.00512	0.58	1.73	43					0.00368	0.57	1.74	43
	Protein import												
XP_629204.1	Stress-70 protein, mitochondrial (mtHsp70)	0.0004	0.6	1.67	62					0.00006	0.53	1.88	62
XP_645160.1	mitochondrial substrate carrier family protein Q	0.00739	3.11	3.11	10					0.00191	2.25	2.25	10
	Other known processes FA metabolism												
XP_645587.1	acetyl-CoA C-acetyltransferase	0.00015	0.42	2.41	37					0.00549	0.61	1.65	35
XP_643323.1	acyl-CoA oxidase A	0.01325	2.16	2.16	25					0.00006	5.54	5.54	25
XP_635485.1	electron transfer flavoprotein alpha subunit	0.01932	0.51	1.97	24					0.00615	0.42	2.36	23
XP_642058.1	electron transfer flavoprotein beta subunit	0.03524	0.6	1.65	21					0.022	0.54	1.86	21
XP_645815.1	Redox superoxide dismutase/SOD	0.01603	0.46	2.18	10					0.00404	0.4	2.52	12
XP_639996.1	Uncharacterized proteins putative mitochondrial transferase caf17	0.00542	0.41	2.41	24					0.00006	0.28	3.59	24
XP_639145.1	putative delta-1-pyrroline-5-carboxylate dehydrogenase	0.04023	0.63	1.59	38					0.00006	0.49	2.04	37

## Data Availability

LC-MS data are available on request at the Mass Spectrometry Laboratory IBB PAS.

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
