# Peer review of "Mitochondrial Processes during Early Development of Dictyostelium discoideum: From Bioenergetic to Proteomic Studies"

_genes, 2021, doi:10.3390/genes12050638_

Round 1

Reviewer 1 Report

The manuscript addresses important questions about energy states during amoeba development. The authors applied a proteomic approach to elucidate the mitochondrial changes. The analysis of mitochondrial proteomes is undoubtedly challenging, however, some of the conclusions in the manuscript are not supported by the data. For example, the authors state that they combined D. discoideum mitochondrial proteomics and functional studies. However, this would require a confirmation that key mitochondrial processes identified by proteomics are altered. This can be achieved, for example, by enzymatic activity assays or measurements of metabolites.  

The title needs to be changed, mitochondria can not be "developed"
In the Methods part: sometimes "micro-" shown as u sometimes as μ. This has to be tidied up. 
I would also suggest presenting proteomic data schematically. This would help the reader to grasp what processes are changed in amoeba during different stages. This can be achieved by using mapping tools like KEGG.

line 37: the phrase "displays the most sophisticated form of multicellularity" not clear. Perhaps higher eukaryotes are "more sophisticated" in terms of multicellularity
line 93: "mitochondria isolation" twice
from line 102: it would be great to present respiratory states/stages as a scheme where all substrates and inhibitors are indicated. Also, the calculations such as "State 3 was calculated by subtracting state 4 from basal respiration" should be shown as a formula. 
line 193:  can "decrease in the rate of state 3" also mean the decreased amount of ATP? At stages A and S, ATP is converted into cAMP and thus the number of ATP molecules will be reduced
Figure 2E: Representative image showing oxygen consumption is not clear. What stage of D. discoideum is presented here? would be good to compare graphs obtained while testing different stages.
Figure 3: Each functional category has to be assigned to the same color. The purpose of this figure is not clear. "protein degradation and biosynthesis" category needs to be separated.
line 286: it is not possible to state that glycolysis was not upregulated as the proteomic analysis was done on mitochondria and many glycolytic enzymes are in the cytosol

Author Response

We are grateful to the Reviewer for all valuable comments and suggestions. 

The revised manuscript have been corrected according to the points raised by the Reviewer. We corrected the title of the manuscript according to the Reviewers suggestions to: “Mitochondrial Processes During Early Development of Dictyostelium discoideum: From Bioenergetic to Proteomic Studies”. All suggestion was taken into account and was resulted in following corrections:

The manuscript addresses important questions about energy states during amoeba development. The authors applied a proteomic approach to elucidate the mitochondrial changes. The analysis of mitochondrial proteomes is undoubtedly challenging, however, some of the conclusions in the manuscript are not supported by the data. For example, the authors state that they combined D. discoideum mitochondrial proteomics and functional studies. However, this would require a confirmation that key mitochondrial processes identified by proteomics are altered. This can be achieved, for example, by enzymatic activity assays or measurements of metabolites.

We appreciate Reviewer suggestions regarding enzymatic activity assays or measurements of metabolites. However except of the proteomic adata we present bioenergetics therefore accorging to the Rewievier suggestion we changed in title „functional studies to bioenergetic” and consequently

-line 23 and 81

From

„funtional”

to

„bioenergetic”

The title needs to be changed, mitochondria can not be "developed"

We have changed the title

From

„The early development of the mitochondria of Dictyostelium discoideum: From Functional to Proteomic Studies.”

to:

„Mitochondrial Processes During Early Development of Dictyostelium discoideum: From Bioenergetic to Proteomic Studies”.

In the Methods part: sometimes "micro-" shown as u sometimes as μ. This has to be tidied up.

It was corrected, we are sorry for this mistakes.

I would also suggest presenting proteomic data schematically. This would help the reader to grasp what processes are changed in amoeba during different stages. This can be achieved by using mapping tools like KEGG.

We agree with the Reviewer and according to this suggestion. We add Figure 4C as a summary of protemic analysis. Figure 4C is a diagram of downregulated (blue ) and upregulated (orange) mitochondrial proteins in D. discoideum early development as a consequence of starvation process. For clarity of the diagram we introduces of the proteins: idhB; isocitrate dehydrogenase (NAD+); scsA; GTP-specific succinyl-CoA synthetas and mdhB; malate dehydrogenase whci were also added into Table 1.

The Figure 4C was cited in lines: 318,336,365,431.

Figure 4 caption is as follows:

-line 483

From:

„The mitochondrial proteins upregulated and downregulated in stage S when compared to stage U. Data were obtained from qualitative comparative analysis of the D. discoideum MitoProteom of the early developmental stages”

To”

The mitochondrial proteins upregulated and downregulated during D. discoideum starvation. A- upregulated mitochondrial proteins (orange); B. downregulated mitochondrial proteins (blue). Data were calculated from the ratio values obtained from qualitative comparative analysis of the D. discoideum for U/S stages see Methods 2.5. C- diagram  of downregulated (blue)  and upregulated (orange)  mitochondrial proteins in D. discoideum early development as a consequence of starvation process. D. discoideum cells during starvation adapt metabolically to aggregate under glucose limitation and downregulate TCA, Fatty acids beta oxidation processes and enter gluconeogenesis. The early steps of gluconeogenesis is regulated by mitochondrial PCK2 (phosphoenolpyruvate carboxykinase) which allows for alternative glucose/biosynthesis intermediates required for cell aggregation. PEP- (phosphoenolopyruvate); OM- mitochondrial outer membrane, IM-mitochondrial inner membrane”.

line 37: the phrase "displays the most sophisticated form of multicellularity" not clear. Perhaps higher eukaryotes are "more sophisticated" in terms of multicellularity

„most sophisticated” was removed

-line 93: "mitochondria isolation" twice:

"mitochondria isolation" was removed

from line 102: it would be great to present respiratory states/stages as a scheme where all substrates and inhibitors are indicated. Also, the calculations such as "State 3 was calculated by subtracting state 4 from basal respiration" should be shown as a formula. 

According to Reviewer suggestion we have added this kind of formula to Figure 2. For clarity the numbers of Figure were changed. Fig. 2E is now Fig. 2A. In this figure we added representative traces for U, A and S stages and introduced all substances and inhibitors. The actual Figure 2B is the previous Fig. 2A, the actual Figure 2C is the previous Fig. 2B, the actual Figure 2D is previous Fig. 2C and the actual Fig. 2E is previous Fig. 2D. For citation of the Figure 2A and Supplementary file S1, we introduced new sentences in the text:

-line 215-218

“The representative traces of U, A and S cells were presented in the Figure 2A where the exact substances and inhibitors were introduced. The raw data taken for the calculation of all bioenergetic parameters are presented in Supplementary file S1.

line 193:  can "decrease in the rate of state 3" also mean the decreased amount of ATP? At stages A and S, ATP is converted into cAMP and thus the number of ATP molecules will be reduced.

We think that it is difficult to address ATP level only at the basis of state 3 rate. Lower rate of state 3 can result in lower ATP level but also can co-occurred with high ATP level. Moreover, as shown by Hiraoka et al, 2019, both ATP rich and ATP-poor cells maintain their ATP levels during the aggregation process from U stage to the mound stage, and this intracellular ATP levels influence these cell fates. Therefore in our suggestions we only address ATP synthesis.

Figure 2E: Representative image showing oxygen consumption is not clear. What stage of D. discoideum is presented here? would be good to compare graphs obtained while testing different stages.

This figure was changed what we have already explained.

Figure 3: Each functional category has to be assigned to the same color. The purpose of this figure is not clear. "protein degradation and biosynthesis" category needs to be separated.

We have changed colors in Figure 3 according to the Reviewer suggestion. The presence of the same group for „protein of synthesis and degradation” is based on the same methodology as for group named „fatty acids metabolism including fatty acids synthesis and degradation. Consequently we have changed the name of this group to „Protein metabolism”.

-line 337, It was changed the captions 3.3.2

from

„Protein biosynthesis and degradation”

To

„Protein metabolism”

 -line 286: it is not possible to state that glycolysis was not upregulated as the proteomic analysis was done on mitochondria and many glycolytic enzymes are in the cytosol.

We agree with a Reviewer, we have added the sentence:

-line 323-325: “This is because the analysed mitochondria proteins are not in the cytosol where glycolytic enzymes are present”.

Reviewer 2 Report

The Early Development of the Mitochondria of Dictyostelium discoideum: From Functional to Proteomic Studies

Monika Masur et al.

In their manuscript “The early development of the mitochondria of Dictyostelium discoideum: From functional to proteomic studies “, Monika Masur et al. reveal, by functional and proteomic approaches focusing on mitochondrial metabolism, significant differences between the early developmental stages of Dictyostelium discoideum (unicellular, aggregation and streams) induced by starvation. They suggest that key mitochondrial processes are downregulated during early development. The article is well written, clear and the results seem convincing. However, it would be very interesting to have in addition to the respiratory activities and the proteomic study some information on the mitochondrial structure (rhodamine labelling for example) and the state of OXPHOS by native gels approaches.

Minor point:

Line 225, supplementary data Table 1 should be extended with more information, such as gene name, number of identified peptides, enrichment factor, peptide sequence and a brief description. This table has no interest in the current state.

Line 286, how can you get data on glycolysis with proteomic analysis on isolated mitochondria. The authors should modify this sentence. However, this question is very interesting and could have been addressed by a proteomic analysis on the whole cell.

Line 399, the authors propose that ROS play a role as a signalling molecule.  The authors could also have simply considered that mitochondrial metabolism reprogramming can lead to an increased ROS production.

Line 424, the authors could indicate in the graph (E) below the curve the different states they are measuring (state 4, Basal and maximal).

Author Response

We are grateful to the Reviewer for all valuable comments and suggestions. The revised manuscript have been corrected according to the points raised by the Reviewer. All suggestion was taken into account and was resulted in following corrections:

In their manuscript “The early development of the mitochondria of Dictyostelium discoideum: From functional to proteomic studies “, Monika Masur et al. reveal, by functional and proteomic approaches focusing on mitochondrial metabolism, significant differences between the early developmental stages of Dictyostelium discoideum (unicellular, aggregation and streams) induced by starvation. They suggest that key mitochondrial processes are downregulated during early development. The article is well written, clear and the results seem convincing. However, it would be very interesting to have in addition to the respiratory activities and the proteomic study some information on the mitochondrial structure (rhodamine labelling for example) and the state of OXPHOS by native gels approaches.

We was observing the mitotracker stained mitochondria by confocal microscope and did not see any spectacular changes in their structure. Unfortunately, the valuable rhodamine labelling, addition state of OXPHOS by native gels approaches are not possible to be done by us at this moment. We consider this valuable suggestions in the future. It is not possible to start D. discoideum cell culture and perfome mitochondria purification in few days.

Minor point:

Line 225, supplementary data Table 1 should be extended with more information, such as gene name, number of identified peptides, enrichment factor, peptide sequence and a brief description. This table has no interest in the current state.

We agree with a Reviewer. This Supplementary file was changed. According to the Reviewer suggestion we enclosed additional file into this supplement, we introduced: Protein description (Peptide sequence); calculated mass; protein score, experimental mass; number of identified peptides, identified charge states; peptide score. It was also changed the Supplementary files:

From „Supplementary data 1” presenting identified mitochondrial proteins enriched in LC-MS MS analysis

to Supplementary file S2 and

„Supplemenatry data Table 2

To

Supplemenary file S1” presenting original (raw) data of oxygen consumption in intact cells.

 -line 217-218, the sentence was added

“The raw data taken for the calculation of bioenergetic parameters are presented in Supplementary file S1”.

-line 468

From

Supplemenatry data Table 2

to

Supplementary file S1

Line 286, how can you get data on glycolysis with proteomic analysis on isolated mitochondria. The authors should modify this sentence. However, this question is very interesting and could have been addressed by a proteomic analysis on the whole cell.

We agree with a Reviewer, the sentence was added

-line 323-325

“This is because the analysed mitochondria proteins are not in the cytosol where glycolytic enzymes are present”.

Line 399, the authors propose that ROS play a role as a signalling molecule. The authors could also have simply considered that mitochondrial metabolism reprogramming can lead to an increased ROS production.

We agree with the Reviewer. The sentence was added.

-line 439-441: „On the other hand ROS production could simply increase as a response of mitochondrial metabolism reprogramming. Thus the additional studies in this field are needed”.  

 Line 424, the authors could indicate in the graph (E) below the curve the different states they are measuring (state 4, Basal and maximal).

According to Reviewer suggestion we add this kind of formulas to Figure 2E. Figure 2E was changed to Figure 2A, and order was consequently changed. For clarity, in the actual Fig. 2A we also added representative traces for A and S stages and introduced all substances and inhibitors.

Reviewer 3 Report

Brief summary

The manuscript by Mazur et al. reports changes in the mitochondrial functions and mitochondrial proteome during Dictyostelium discoideum cell development, which could be associated with adaptation to starvation. Although there is much still to be understood about how D. discoideum cells adapt to starvation, the study allowed the identification of proteins and potential pathways involved. A few issues need to be considered before publication.

Comments

  1. Dictyostelium discoideum cells stop proliferating during starvation, the authors describe that 2x10^6 cells were taken to measure cellular respiration per condition. Increased mitochondrial respiration in vegetative cells could rise due to a high number of mitochondria per cell, and this could describe the alteration in the levels of mitochondrial proteins. The authors could check mitochondrial count per cell in cells from U, A, and S stages.

  1. The authors should clarify the “Down in A and Up in S” tendency in the method section. It is not clear how Table 1 and Figure 4 are correlated. Table 1 shows ratio and folds changes for each comparative study. It will be better to write a bit more on how the calculation is done to get the data in Figure 4.

  1. Authors should rewrite the sentence in Line 51-53, which describes the early developmental stages.

  1. The authors should include a reference for the last statement in Line 53.

  1. The authors should rewrite to explain cell culture and mitochondrial isolation techniques that were adopted from reference 14, in Lines 93-94.

  1. In Line 196, Figure 2B should be 2C and vice versa.

  1. Supplementary data table 1 comes in line 227 after table 2 in figure 2 legend. Order of their appearance should be taken into consideration. Excel tab name of Supplementary table 1 is not in English. Supplementary table 2 tab named Row data should be “Raw data”.

  1. In Line 120, the authors need to describe or indicate a reference more clearly for the preparation of mass spectrometry protein samples from cells.

  1. Line 194 sentence for Figure 2C should be re-written to make it clear to understand.

  1. The authors need to include/cite Figure 2E in the text.

  1. Reference 32 in line 279 should be placed after the previous sentence in line 277, which describes gluconeogenesis.

  1. The manuscript’s title is a bit confusing and could benefit from some changes. The authors showed that mitochondrial function is downregulated during early development. The suggested title “Mitochondrial processes during early development of Dictyostelium discoideum: From functional to proteomic studies”. Need to rewrite the title again.

  1. Abbreviation of Dictyostelium discoideum in parenthesis is not necessary, after all when you repeat the name, you write the abbreviated form D. discoideum.

Take or leave suggestions: D. discoideum cells start reprogramming themselves in response to starvation within 6 hours. The article would benefit from measurement of mitochondrial functions in cells starved for 6 and 12 hours, which may explain more about adaptation to early starvation.

Author Response

We are grateful to the Reviewer for all valuable comments and suggestions. The revised manuscript have been corrected according to the points raised by the Reviewer. We corrected the title of the manuscript according to the Reviewers suggestions to: “Mitochondrial Processes During Early Development of Dictyostelium discoideum: From Bioenergetic to Proteomic Studies”. All suggestion was taken into account and was resulted in following corrections:

The manuscript by Mazur et al. reports changes in the mitochondrial functions and mitochondrial proteome during Dictyostelium discoideum cell development, which could be associated with adaptation to starvation. Although there is much still to be understood about how D. discoideum cells adapt to starvation, the study allowed the identification of proteins and potential pathways involved. A few issues need to be considered before publication.

Comments

  1. Dictyostelium discoideumcells stop proliferating during starvation, the authors describe that 2x10^6 cells were taken to measure cellular respiration per condition. Increased mitochondrial respiration in vegetative cells could rise due to a high number of mitochondria per cell, and this could describe the alteration in the levels of mitochondrial proteins. The authors could check mitochondrial count per cell in cells from U, A, and S stages.

We agree with the Reviewer that cells in stages U, A and S contain different amount of mitochondria. This is reflected in different yield of mitochondria isolation procedure - we added the information at the end of the part 2.1 of the Methods.

-line 97-112:

“For mitochondria isolation cell cultures (850 ml)at a given stage were centrifuged at 600×g for 5 min and then washed twice in phosphate medium A (14.5 mM KH2PO4, 5 mM Na2HPO4, pH 6) by centrifugation at 600×g for 5 min. Next, the cells were homogenized in medium B (0.38 M sucrose, 20 mM Tris-Cl, 0.5 mM EDTA, 1% defatted bovine serum albumin [BSA], pH 7.5) using glass/Teflon homogenizer and centrifuged at 860×g for 5 min. After this step supernatant was centrifuged at 10 000×g for 15 min and obtained pellet was suspended in medium C (0.38 M sucrose, 20 mM Tris-Cl, 0.5 mM EDTA, 0.6% defatted BSA, pH 7.2), and centrifuged at 860×g for 5 min to remove debris. Supernatant was centrifuged at 10 000×g for 10 min, mitochondrial pellet suspended in medium D (0.38 M sucrose; 20 mM, Tris-Cl, pH 7.2) and centrifuged at 10 000×g for 15 min. Mitochondrial pellet was suspended in medium D and used for further analysis. All centrifugation procedures were carried out at 4°C and media used for mitochondria isolation were ice cold. The calculated mean values of the yield of mitochondria for 1 g of cells were 5 mg, 1.4 mg and 1.7 mg for stage U, A and S, respectively”.

Consequently, to avoid the problem with the different amount of mitochondria, for bioenergetics analysis we used parameters based on the comparison of rates determined for a given cell stage. We added the information to the part 3.1 of the Results and introduced appropriate changes later in this part. We also agree that we could additionally check mitochondrial count per cell in cells from U, A, and S stages. However, it is not possible to start D. discoideum cell culture for mitochondria counting within 7 days given for the manuscript revision.

As far as the levels of mitochondrial proteins is concerned we could not agree with the Reviewer as the comparative proteomic analysis was performed for the same amount of isolated mitochondria. We are sorry for the lack of the relevant information in the part 2. 3 of the Methods describing sample preparation for MS analysis. This was corrected.

-line 139-141:

From:

“Proteins were suspended in a dissolution buffer (8 M of urea, 2% CHAPS, 10 mM of HEPES, with a pH of 8.5)”.  

To

“5 ug of isolated mitochondria were suspended in a dissolution buffer (8 M of urea, 2% CHAPS, 10 mM of HEPES, with a pH of 8.5)”.   

-line 212-215: the sentence was added:

“The parameter calculation allows for comparison of cells containing different amount of mitochondria, which was observed herein (Methods, p. 2.1) and has already been reported for cell in stages U, A and S [8]”

-line 218-221 the sentence was added:

“The representative traces registered for cells in stages U, A and S are presented in the Figure 2A whereas raw data used for the calculation of all bioenergetic parameters are presented in Supplementary file S1.”

-line 223-225: the sentence was added:

“However, according to the yield of mitochondria isolation it could be assumed that cells in stages A and S contained the comparable mitochondria mass that was lower than in cell in U stage”.

-line 200-203 the sentence was changed.

from

-line 229-232:

From

“Accordingly, mitochondrial coupling efficiency corresponding to state 3 respiration contributed to basal respiration decreasing to the same level for the cells in stages A and S when compared to the cells in stage U (Figure 2B)”.

to

“Accordingly, mitochondrial coupling efficiency corresponding to state 3 respiration contribution to basal respiration decreased to the same level for the cells in stages A and S when compared to the cells in stage U (Figure 2C)”.

-line 233-236, the sentence was corrected:

From

“However, the cells in stages A and S differed in their rates of maximal respiration, i.e., the uncoupling capacity of FCCP (Figure 2A), which resulted in differences in mitochondrial coupling capacity (Figure 2C), defined as the ratio of maximal respiration to state 4 respiration”.

To:

“However, the cells in stages A and S differed in the uncoupling capacity of FCCP (Figure 2B), which resulted in differences in mitochondrial coupling capacity (Figure 2D),defined as the ratio of maximal respiration to state 4 respiration”.

-line 202-203, the sentence was corrected:

from

“The respiratory state of mitochondria is a dynamic state resulting from crosstalk between the 2 bioenergetic states of state 3 and state 4 [16,19]”.

to

“The respiratory state of mitochondria is a dynamic state resulting from crosstalk between the two bioenergetic states termed state 3 and state 4 [16,19]”.

- line 237-239, the sentence was removed:

“Since state 4 respiration was comparable in the studied cells, we could assumed that there were differences in substrates oxidation”.

-line 242-244, the sentence was corrected:

from

“The parameter corresponded to the difference between maximal respiration and basal respiration and reflected how close it was operating to the cell's bioenergetic limit [19]”

to

“The parameter corresponds to the difference between maximal respiration and basal respiration and reflects how close to its bioenergetic limit a cell is operating [19]”.

-Line 464-478, Figure 2 caption was changed.

From

“Figure 2. The functional state of mitochondria in intact D. discoideum cells in the studied stages. Aoxygen consumption rates; B—coupling capacity (maximal respiration to state 4 ratio); C—coupling efficiency (contribution of state 3 to basal respiration); D—spare respiratory capacity (the difference between maximal respiration and basal respiration); E—a representative trace applied for calculations. The rate of oxygen uptake by cells (2 × 106) was measured in 0.5 ml of DB in a water-thermostated incubation chamber with a computer-controlled Clark-type O2 electrode at 18°C (Oxygraph, Hansatech, UK). The data are presented as mean values ± SD of 3 independent experiments. *—p < 0.05; **—p < 0.005, **—p < 0.001. Calculated original data are presented in Supplementary data table 2”.

To

„Figure 2.The functional state of mitochondria in intact D. discoideum cells in the studied stages. A—representative traces for cells in studied stages applied for calculations as well as explanation of the calculated parameters ; Boxygen consumption rates under the applied oxygen uptake rate measurement condition; C—coupling efficiency (contribution of state 3 to basal respiration); D—coupling capacity (maximal respiration to state 4 ratio); E—spare respiratory capacity (the difference between maximal respiration and basal respiration). The rate of oxygen uptake by cells (2 × 106) was measured in 0.5 ml of DB in a water-thermostated incubation chamber with a computer-controlled Clark-type O2 electrode at 18°C (Oxygraph, Hansatech, UK). Tributyltin (FCCP) was add in concentration of (0.5–1.5 uM) to enforced the state 4. The maximal respiration was induced by using (0.1–0.24) uM of carbonyl cyanidep-trifluoromethoxyphenylhydrazone, (FCCP); sodium dithionite is a zero oxygen solution powder used for calibration of oxygen sensors at zero oxygen.The data are presented as mean values ± SD of 3 independent experiments. *—p < 0.05; **—p < 0.005, **—p < 0.001. Calculated original data are presented in Supplementary file S1.

2. The authors should clarify the “Down in A and Up in S” tendency in the method section. It is not clear how Table 1 and Figure 4 are correlated. Table 1 shows ratio and folds changes for each comparative study. It will be better to write a bit more on how the calculation is done to get the data in Figure 4.

The addition information have been added to the Methods 2.5:

- line 189-198:

The tendency in comparative studies as down in A and Up in S, was calculated from the ratio values of comparative qualitative studies for phase to phase ratio quotient U/A and A/S (Table 1). The expression profile was calculated against U state as a basal of expression level (1). The ratio value (>1) and (<1) was regarded as upregulated and downregulated respectively. The phase to phase ratio quotient [S/U]' was taken for proteins: isocitrate dehydrogenase (NAD+), cytochrome c oxidase subunit IV and heat shock protein Hsp70 family protein. For other proteins the [S/U], was calculated from equation: [A/U]/[A/S], where [A/U] and [A/S] were given as the ratio in section 'Phase to phase ratio quotient [A/U]' and section 'Phase to phase ratio quotient [A/S]'”.

 -line 498-511

From:

Figure 4. The mitochondrial proteins upregulated and downregulated in stage S when compared to stage U. Data were obtained from qualitative comparative analysis of the D. discoideum MitoProteom of the early developmental stages”

To”

“Figure 4. The mitochondrial proteins upregulated and downregulated in stage S when compared to stage U during D. discoideum early starvation. A- upregulated mitochondrial proteins; B. downregulated mitochondrial proteins. The protein upregulation and downregulation were estimated using qualitative comparative analysis between stages U and S (see Methods 2.5). C- diagram of downregulated (blue) and upregulated (orange) mitochondrial proteins assigned to cellular processes. The cell metabolic adaptation under glucose depletion includes attenuation of Krebs cycle, Fatty acids beta oxidation and oxidative phosphorylation, and enhancement of amino acids’ metabolism being known fuel for gluconeogenesis. The early steps of gluconeogenesis is regulated by mitochondrial phosphoenolpyruvate carboxykinase (PCK2) which allows for alternative glucose/biosynthesis intermediates required for cell aggregation. TCA (tricarboxylic acid cycle, Krebs cycle); PEP (phosphoenolopyruvate); SOD (superoxide dismutase); caf17 (putative mitochondrial transferase); scsA (succinate-CoA ligase); mdhB (mitochondrial malate dehydrogenase); idhB (isocitrate dehydrogenase (NAD+)); α-KG (α- ketoglutarate); OM (mitochondrial outer membrane); IM (mitochondrial inner membrane)”.

In the Table 1 we introduced short names of 4 proteins:

For Isocitrate dehydrogenase (NAD+) is (idhB);

succinate-CoA ligase is (scsA)

malate dehydrogenase is (mdhB)

phosphoenolpyruvate carboxykinase is (PCK2)

3. Authors should rewrite the sentence in Line 51-53, which describes the early developmental stages.

  We are sorry for this, the sentence was corrected.

  -line 54-56:

  From  

  „D. discoideum cells, during the early steps of aggregation when unicellular cells start to move and form waves, also referred to as early developmental stages induced by starvation (Figure 1) (Schilde and Schaap, 2013)”.

  to:

 „D. discoideum cells, during the early steps of aggregation when unicellular cells start to move  and form waves, are referred to as early developmental stages induced by starvation (Figure 1) (Schilde and Schaap, 2013).

4. The authors should include a reference for the last statement in Line 53.

-line 56-57:

From

“These early developmental stages have been studied previously”.

to

“These early developmental stages have been studied previously[7,8]”.

5. The authors should rewrite to explain cell culture and mitochondrial isolation techniques that were adopted from reference 14, in Lines 93-94.

We included the procedure of mitochondria isolation to Methods 2.1

-line 97-111:

“For mitochondria isolation cell cultures (850 ml)from U, A and S stages were centrifuged at 600×g for 5 min and then washed twice in phosphate medium A (14.5 mM KH2PO4, 5 mM Na2HPO4, pH 6) by centrifugation at 600×g for 5 min. Next, the cells were homogenized in medium B (0.38 M sucrose, 20 mM Tris-Cl, 0.5 mM EDTA, 1% defatted bovine serum albumin [BSA], pH 7.5) using glass/Teflon homogenizer and centrifuged at 860×g for 5 min. After this step supernatant was centrifuged at 10 000×g for 15 min and obtained pellet was suspended in medium C (0.38 M sucrose, 20 mM Tris-Cl, 0.5 mM EDTA, 0.6% defatted BSA, pH 7.2), and centrifuged at 860×g for 5 min to remove remaining cells. Supernatant was centrifuged at 10 000×g for 10 min, mitochondrial pellet suspended in medium D (0.38 M sucrose; 20 mM, Tris-Cl, pH 7.2) and centrifuged at 10 000×g for 15 min. Mitochondrial pellet was suspended in medium D and used for further analysis. All centrifugation procedures were carried out at 4°C and media used for mitochondria isolation were ice cold and. The calculated mean values of the yield of mitochondria for 1 g of cells were 5 mg, 1.4 mg and 1.7 mg for stage U, A and S, respectively”.

6. In Line 196, Figure 2B should be 2C and vice versa.

In Figure 2, we have replaced 2B. with 2C. We are sorry for this mistake.

Finally we have changed the order in Figure 2. Figure 2E was changed to Figure 2A, and order of the other was consequently changed. 

7. Supplementary data table 1 comes in line 227 after table 2 in figure 2 legend. Order of their appearance should be taken into consideration. Excel lab name of Supplementary table 1 is not in English. Supplementary table 2 tab named Row data should be “Raw data”.

We agree with a Reviewer. The order of Supplementary files was changed.  Also the name of files was changed as follows:

Supplementary data Table 1 and 2 to Supplementary file S1 and S2.

Moreover according to Reviewer suggestion we enclosed additional data into Supplementary file S2,  regarding identified proteins: Protein description (Peptide sequence); calculated mass; protein score, experimental mass; number of identified peptides, identified charge states; peptide score.

-line 218-219 the sentence was added

“The raw data taken for the calculation of bioenergetic parameters are presented in Supplementary file S1”.

8. In Line 120, the authors need to describe or indicate a reference more clearly for the preparation of mass spectrometry protein samples from cells.

The digestion procedure is written in the text. We have added some correction to make it more clear.

-line 138:

from:

„Proteins were suspended in a dissolution buffer (8 M of urea, 2% CHAPS, 10 mM of HEPES, with a pH of 8.5).

To

“5 ug of purified mitochondria were suspended in a dissolution buffer (8 M of urea, 2% CHAPS, 10 mM of HEPES, with a pH of 8.5)”.

-line 140-142:

from

“Samples were using modified the single-pot solid-phase-enhanced sample preparation (SP3) with some modifications [20]. The exact sample preparation protocol was as follows”.

to:

“Samples were processed using modified the single-pot solid-phase-enhanced sample preparation (SP3) method [20]. The exact sample preparation protocol was as follows”.

9. Line 194 sentence for Figure 2C should be re-written to make it clear to understand.

We are sorry for the mistake in the Figure 2 assemble. The mistake was corrected and we hope the mentioned sentence is clear now.

-Line 230-233

From:

“Accordingly, mitochondrial coupling efficiency corresponding to state 3 respiration contributed to basal respiration decreasing to the same level for the cells in stages A and S when compared to the cells in stage U (Figure 2B).

To

“Accordingly, mitochondrial coupling efficiency corresponding to state 3 respiration contribution to basal respiration decreaed to the same level for the cells in stages A and S when compared to the cells in stage U (Figure 2C)”.

10. The authors need to include/cite Figure 2E in the text.

We are sorry for this mistake. The Figure 2E – the actual Figure 4A was included in the text:

-line 216-217, the sentence was added:

“The representative traces of U, A and S cells were presented in the Figure 2A where the exact substances and inhibitors were introduced.”

11. Reference 32 in line 279 should be placed after the previous sentence in line 277, which describes gluconeogenesis.

 We agree with the Reviewer, the citation was replaced.

12. The manuscript’s title is a bit confusing and could benefit from some changes. The authors showed that mitochondrial function is downregulated during early development. The suggested title “Mitochondrial processes during early development of Dictyostelium discoideum: From functional to proteomic studies”. Need to rewrite the title again.

We agree with a Reviewer and changed the title of the manuscript to:

„Mitochondrial processes during early development of Dictyostelium discoideum: From bioenergetic to proteomic studies”

13. Abbreviation of Dictyostelium discoideum in parenthesis is not necessary, after all when you repeat the name, you write the abbreviated form  discoideum.

We have changed this in the following lines:

- line 18,

from

“The slime mould Dictyostelium discoideum’s (D. discoideum) life cycle includes different..)

to

 “The slime mould Dictyostelium discoideum’s life cycle includes different..)

-line 39

from

“The unicellular slime mould Dictyostelium discoideum (D.discoideum), one of the 150…”

to

“The unicellular slime mould Dictyostelium discoideum, one of the 150…”

Take or leave suggestions: D. discoideum cells start reprogramming themselves in response to starvation within 6 hours. The article would benefit from measurement of mitochondrial functions in cells starved for 6 and 12 hours, which may explain more about adaptation to early starvation.

Thank you for this valuable suggestion, we will take it into account in our next studies.